# Can Reduced Intake Associated with Downsizing a High Energy Dense Meal Item be Offset by Increased Vegetable Variety in 3–5-year-old Children?

**DOI:** 10.3390/nu10121879

**Published:** 2018-12-03

**Authors:** Sharon A. Carstairs, Samantha J. Caton, Pam Blundell-Birtill, Barbara J. Rolls, Marion M. Hetherington, Joanne E. Cecil

**Affiliations:** 1Population and Behavioral Sciences, School of Medicine, University of St Andrews, St Andrews KY16 9TF, UK; jc100@st-andrews.ac.uk; 2School of Health and Related Research (ScHARR), University of Sheffield, Sheffield S1 4DA, UK; s.caton@sheffield.ac.uk; 3School of Psychology, University of Leeds, Leeds LS2 9JT, UK; p.birtill@leeds.ac.uk (P.B.-B.); m.hetherington@leeds.ac.uk (M.M.H.); 4Department of Nutritional Sciences, College of Health and Human Development, Pennsylvania State University, University Park, PA 16802, USA; bjr4@psu.edu

**Keywords:** portion size, pre-school children, eating behavior, variety

## Abstract

Large portions of energy dense foods promote overconsumption but offering small portions might lead to compensatory intake of other foods. Offering a variety of vegetables could help promote vegetable intake and offset the effect of reducing the portion size (PS) of a high energy dense (HED) food. Therefore, we tested the effect on intake of reducing the PS of a HED unit lunch item while varying the variety of the accompanying low energy dense (LED) vegetables. In a within-subjects design, 43 3–5-year-old pre-schoolers were served a lunch meal in their nursery on 8 occasions. Children were served a standard (100%) or downsized (60%) portion of a HED sandwich with a side of LED vegetables offered as a single (carrot, cherry tomato, cucumber) or variety (all 3 types) item. Reducing the PS of a HED sandwich reduced sandwich (g) (*p* < 0.001) and total meal intake (kcal) consumption (*p* = 0.001) without an increased intake of other foods in the meal (LED vegetables (*p* = 0.169); dessert (*p* = 0.835)). Offering a variety of vegetables, compared with a single vegetable, increased vegetable intake (g) (*p* = 0.003) across PS conditions. Downsizing and variety were effective strategies individually for altering pre-schoolers’ intakes of HED and LED meal items, however, using variety to offset HED downsizing was not supported in the present study.

## 1. Introduction

The portion size effect (PSE), where more is eaten when large portions are offered compared with small portions, is robust in adult [1,2] and child populations [3,4,5]. There is evidence to suggest that susceptibility to the PSE is influenced by individual differences, such as liking [6], sex [7], body size [8], and eating traits [5,9]; however, these findings have not been consistently replicated. The PSE is associated with a sustained increase in energy intake over several days [2,10], and without energy compensation [11]. Given that portion sizes of some energy dense foods have increased over the years [12,13], a lack of compensatory behaviors to increased portion size may promote overeating and excessive energy intakes. In contrast, small portions might affect compensatory behaviors to offset any portion downsizing. In the present study, we tested the effect of reducing the portion size of a HED item on lunch intake and tested whether offering a variety of vegetables increased vegetable intake and offset reduced energy intake associated with this downsizing.

Downsizing portions of energy dense (ED) foods and its effect on consumption has previously been investigated with mixed findings [8,14,15,16,17,18]. Studies conducted in children showed that both 50% [14] and a 25% reduction [8] in entrée portion significantly reduced entrée intake. However, another similar study in pre-school children found no change in entrée intake following a 25% portion reduction [16]. Furthermore, the impact of reducing portion sizes of HED entrées on consumption of other foods within the meal, including vegetables, show varied results [8,14,16]. Sensory cues play an important role in overall food consumption [19] and these different findings may be influenced by food pairing, such that altering one food may change the acceptance of others on the plate [20]. Thus, the variety of foods and flavors offered at a meal could play an important role in overall meal intake.

Similar to the PSE, food variety is associated with increased food consumption in adults and adolescents both within controlled laboratory conditions [21,22] and in field studies [23]. The variety effect has been explained by sensory-specific satiety (SSS), where the appeal of the consumed food decreases compared to those foods not consumed [24,25,26]. Variety has been used to facilitate vegetable intake in children [27], which is an important observation given the low vegetable intakes evident globally [28]. However, these results are not consistent across environments, with studies offering a choice of vegetable variety to children in a restaurant setting [29] and in the home [30] showing little or no effect on vegetable consumption. If variety has the potential to increase intake of low-energy dense (LED) foods such as vegetables it could also be used to offset the effect of downsizing. Offsetting in behavioral science refers to the tendency to compensate for changes in the environment [31]. Thus, providing a small portion of a HED food compared with a larger portion, alongside a variety of vegetables compared with a single vegetable, might produce compensatory behavior, in this case an increased consumption of vegetables. If a small portion is paired with a variety of foods which are low in energy density (e.g., vegetables), then any compensation may still yield a net reduction in energy intake, as well as an increase in vegetable intake, as observed by Savage et al [8].

The aim of this study, therefore, was to investigate the effects of downsizing combined with variety on food intake in pre-school children. We tested the effects of downsizing the portion of an energy dense unit-based lunch item (100% to 60%) on food intake and whether providing a variety of vegetables as an accompaniment increased consumption of this item compared with offering a single vegetable. The current study also explored the influence of child eating behavioral traits on consumption, with the assumption that individual differences might predict the response to downsizing and the effect of variety.

## 2. Materials and Methods

### 2.1. Experimental Design

In a within-subjects design with 8 weekly conditions (Table 1), children were offered a lunch meal at nursery during a normal lunchtime setting. Lunches were either a standard or downsized portion (100%, 60%) of a HED food (>2.5 kcal/g as defined by Albar et al., [32] N.B. HED is alternatively defined as >2.25 kcal/g [33]) (nutritional information shown in Table 2) with a side of LED vegetables offered as a single or variety item. The order of the experimental conditions was counterbalanced by using Latin squares assigned for each nursery group and by alternating the starting portion size block.

### 2.2. Participants

Participants were 3–5-year-old pre-school children, recruited by distributing letters to parents of children in host nurseries within Fife and Tayside (Northeast Scotland). Parents provided written, informed consent for the participation of their child in the study as well as their own participation in completing parental questionnaires. Children who were allergic to any of the foods to be served in the study (identified from screening questionnaire) were excluded from participation. The University of St Andrews School of Medicine Ethics Committee reviewed and approved all procedures for this study (MD12354).

Power calculations based on 80% power to detect a 40 g difference (standardized effect size of 0.5) in intake between two portion conditions at a 5% level of significance revealed that a target of 48 children should be recruited for the study. This estimate is consistent with research demonstrating significant effects of portion size and energy manipulation in young children using a within-subjects design [34,35].

### 2.3. Test Meal and Procedures

The lunch consisted of a cheese sandwich (HED) accompanied by either a single raw vegetable or a variety of 3 (LED) raw vegetables (cucumber, cherry tomatoes, carrots) (Table 2), chosen for the children’s familiarity with these foods [36]. The sandwich and vegetables were cut into uniform pieces with an equal number of units to ensure consistency across portion size and vegetable manipulations; sandwiches were cut into 8 units, vegetables were cut into 18 units (i.e., 18 units of single vegetable or 6 units of each vegetable in the variety condition).

The recommended 40 g portion of fruit and vegetables [37,38] was used to determine the quantity of vegetables offered. Within the single vegetable meal condition, 120 g of vegetables were provided to match the total quantity offered in the variety vegetable condition. The test meal was accompanied by a glass of water (100 mL) followed by the provision of grapes (40 g) and yogurt (120 g) to ensure that the lunch meal was consistent with national government recommendations [37,38].

The 100% portion exceeded age-specific recommendations whereas the 60% portion matched recommended portion sizes for children in this age group [38,39]. A 40% reduction for downsizing was employed in the current experiment based on previous research showing that a 40% portion size reduction resulted in no differences in dietary intake over the whole day compared to control in obese adults [18]. The full test meal for the 100% portion provided 523–532 kcal (dependent on vegetable selection). The 60% portion provided 376–385 kcal, in line with recommendations for a lunch meal for this age category [37] (Table 2).

The test meal was presented to the children during the lunchtime period in the nursery. Children sat in small groups of 2–6 and were advised that “they could eat as much or as little as they liked”. The number of participating children from each of the 9 nurseries ranged from 2 to 9 children. The researchers observed the children during the lunch meal to ensure that children did not share foods and to ensure dropped foods were recovered.

### 2.4. Outcome Measures

#### 2.4.1. Liking, Food Intake, and Anthropometric Assessment

During the familiarisation session (Table 1), children were asked to rate their liking of each of the foods provided in the test meal using cartoon images of faces, a method previously used with children of this age-group [40]. Children were asked whether they thought each food was “yummy”, “just okay” or “yucky”. Liking data was used to establish the three best liked vegetables from a selection of carrot, cherry tomato, cucumber and red pepper; the red pepper was least liked and not offered in the study (Appendix A). The amount of food consumed was calculated as the difference between pre- and post-meal weights, recorded using digital scales (Ohaus-NV511: Parsippany, NJ, USA). Using a portable stadiometer (Seca: Hamburg, Germany), height (cm) was measured to the nearest cm; weight (kg) was measured to the nearest 0.1 kg using a portable digital scale (Leicester SMSSE-0260: Leicester, UK; Seca: Hamburg, Germany).

#### 2.4.2. Parental Questionnaires

Parents of participating children completed questionnaires on general demographic information, eating traits, parental feeding practices and frequency of eating particular foods. The 10-item Food Neophobia Scale [41] was incorporated for measurement of parental food neophobia, and a 6-item version of the Child Food Neophobia Scale [42] was used in this study for its validity for use in pre-school age populations [43,44]. The validated 35-item Child Eating Behavior Questionnaire (CEBQ) [45,46] evaluated 8 subscales related to eating traits of the child: food responsiveness, emotional over- and under-eating, enjoyment of food, desire to drink, satiety responsiveness, slowness in eating, and food fussiness. Parents rated each item on a 5-point scale (1 = never, 2 = rarely, 3 = sometimes, 4 = often, 5 = always). Parents also completed the 49-item Comprehensive Feeding Practices Questionnaire (CFPQ), a validated measure evaluating 12 parental feeding practices subscales including: child control, emotion regulation, encourage balance and variety, environment, food as reward, involvement, modelling, monitoring, pressure, restriction for health and restriction for weight control, and teaching about nutrition [47]. Parents rated items on a 5-point scale (1 = never/disagree, 2 = rarely/slightly disagree, 3 = sometimes/neutral, 4 = mostly/slightly agree, 5 = always/agree). For both CEBQ and CFPQ subscales, a mean score (ranging 1-5) was calculated within a given subscale and used for analyses. Parents were additionally asked to rank the frequency their child self-served themselves food on a 5-point scale (1 = never, 2 = rarely, 3 = sometimes, 4 = often, 5 = always) and complete a food frequency questionnaire (FFQ) [48].

### 2.5. Data Analysis

Analyses were carried out using SPSS (IBM SPSS Statistics v22, Armonk, NY, USA). Repeated measures Analysis of Variance (ANOVA) was conducted to investigate the effect of portion size and vegetable condition on intakes (HED sandwich (g), LED vegetable (g), dessert (g) intakes, and total energy intake (kcal)). Repeated measures ANOVA was conducted to test for any difference in vegetable intakes (g) between individual single vegetable conditions (carrot, cherry tomato and cucumber) in each of the portion conditions. No significant difference in vegetable intakes across each of the 3 types of single vegetable was found within both portion conditions (*p* ≥ 0.294), therefore a mean single vegetable intake was calculated from the 3 vegetable conditions and to compare against variety. Thus, fixed factors included in the final models were HED portion size (100%, 60%) and vegetable condition (single, variety). The Bonferroni method was used to adjust significance levels for multiple pairwise comparisons between means. Finally, one-way ANOVA was conducted to compare individual vegetable intakes for each vegetable type within the variety conditions to explore whether children consumed equally from across all 3 vegetables.

Pearson’s correlation was used for linear bivariate relationships to explore associations between mean intakes, child age and BMI, eating traits and parental feeding practices. Regression analysis, using a stepwise method, was then conducted to investigate which variables predicted HED and LED intakes. Data presented are means ± standard error of the mean. Results were considered statistically significant at *p* < 0.05.

## 3. Results

### 3.1. Participant Characteristics

Fifty-eight responses from parents for their child to participate were received. Following review of screening questionnaire and criteria, 7 were excluded (child did not attend nursery lunchtime sessions on agreed days of testing). Thus, a total of 51 children aged 3–5 years from 9 nursery groups in Fife and Tayside were enrolled in the study from September 2016 to May 2018. Two children withdrew from participation during the course of the study and six were excluded as non-eaters (defined as those who consumed <10% of the smallest HED portion on at least 4 occasions [49]). Intake data were analyzed for 43 children (23 girls and 20 boys). Repeated measures analysis was conducted on a sample of 40 children as 3 children did not complete all 8 experimental conditions. Characteristics of the children are shown in Table 3. Mean child age was 3.9 years; mean child BMI was 16.5 kg/m^2^. In this sample, 74.4% (*n* = 32) of children were categorized as healthy and 25.6% (*n* = 11) were classed with overweight or obesity (sex-specific BMI-for-age [50]). All children classed with overweight or obesity were girls.

### 3.2. Effects of PS and Vegetable Condition on HED Sandwich Intake

A significant effect of portion size on HED sandwich intake (g) was found (F (1,41) = 15.28, *r* = 0.27, *p* < 0.001) (Figure 1). Mean HED sandwich intake in the 60% portion size condition (48.4 ± 2.9 g) was 21% lower (mean difference of 12.9 ± 3.3 g) than in the 100% portion (61.3 ± 4.4 g). There was no main effect of vegetable condition (single vs. variety; F (1,41) = 0.10, *p* = 0.752) on HED intake (54.5 ± 3.2 g and 55.2 ± 3.8 g sandwich intake from single and variety conditions respectively). There was no significant interaction effect of portion size and vegetable condition (*p* = 0.995) indicating vegetable variety did not offset portion downsizing.

### 3.3. Effects of PS and Vegetable Condition on Total Meal Intake

A significant effect of PS was found on total meal energy intake (F (1,41) = 12.2, *r* = 0.23, *p* = 0.001). Mean total intake was 278.0 ± 10.7 kcal in the 60% PS compared with 322.0 ± 16.5 kcal in the 100% PS (Figure 2). This difference equates to 9–12% of the total energy intake recommended for a child of this age at a lunchtime meal. The difference in total meal kcal was driven only by the effect of portion size condition on HED energy intake (F (1,41) = 14.4, *r* = 0.30, *p* < 0.001). No effect of vegetable condition (*p* = 0.877) and no interaction effect of PS and vegetable condition (*p* = 0.590) was evident on total meal energy intake.

### 3.4. Effects of PS and Vegetable Condition on LED Vegetable Intakes

HED portion size condition had no significant effect on intakes of vegetables (100% = 31.5 ± 4.1 g, 60% = 35.1 ± 5.3 g, (F (1,41) = 1.96, *p* = 0.169) (Figure 3). However, a significant effect of vegetable condition (mean single vs variety) on vegetable intake was evident (F (1,41) = 10.05, *r* = 0.20, *p* = 0.003) (Figure 3). Offering a variety of vegetables at the lunch meal resulted in a higher vegetable intake (37.2 ± 5.2 g) compared with the mean single vegetable option (29.5 ± 4.1 g), without an interaction effect with PS condition (F (1,41) = 1.58, *p* = 0.216). One-way ANOVA showed equal intakes from each of the 3 vegetables (carrot, cucumber and cherry tomato) within both variety conditions (100% PS, *p* = 0.406; 60% PS, *p* = 0.401).

### 3.5. Food Intake, Demographics and Trait Eating Behaviors

Based on the 41 parents who completed demographic information, the majority of the parents were white (96%), in employment (90%), 72% had a household income > £40,000 and 61% of mothers and fathers had an undergraduate degree or higher. No significant associations were found between parental demographics (age, BMI, household income, education and employment status) and child intakes (e.g., mother’s age and child mean HED intake *r* = −0.310, *p* = 0.052).

Child and parent food neophobia scores were significantly correlated (*r* = 0.335, *p* = 0.034). Furthermore, child age was inversely correlated with child food neophobia score (*r* = −0.326, *p* = 0.040) and CEBQ food fussiness (*r* = −0.213, *p* = 0.047). Child age was positively correlated with CFPQ modelling (*r* = 0.355, *p* = 0.027) and CFPQ teaching about nutrition (*r* = 0.363, *p* = 0.019). Child BMI was positively correlated with CEBQ emotional overeating score (*r* = 0.362, *p* = 0.022), CEBQ food responsiveness (*r* = 0.311, *p* = 0.048) and CFPQ restriction for health (*r* = 0.314, *p* = 0.046).

#### 3.5.1. HED Sandwich Intake, Demographics and Eating Traits

Positive correlations were found between child age and HED intake in both the 100% (*r* = 0.539, *p* < 0.001) and 60% conditions (*r* = 0.340, *p* = 0.026). No correlations were found between HED intakes across both portion conditions and child BMI (100% HED condition *p* = 0.212, 60% condition *p* = 0.245). Inverse correlations were evident between satiety responsiveness and HED intake in both the 100% (*r* = −0.426, *p* = 0.007) and 60% conditions (*r* = −0.335, *p* = 0.037) (Figure 4).

Linear regression models were constructed with child age and satiety responsiveness scores as predictors of HED intake within each portion size condition (100%, 60%). In the 60% HED portion, only child age significantly contributed to the model with age accounting for 14% of the variance in HED intake (*R*^2^ = 0.143, F = 6.16, *p* = 0.018).

Analysis revealed a strong model where both age and satiety responsiveness significantly predicted intake in the 100% HED condition (*R*^2^ = 0.41, F = 12.7, *p* < 0.001) with age accounting for 34% of the variance and satiety responsiveness accounting for 7%. An increase in age by 1 year predicted an increased intake of HED sandwich by 24.3 g (*p* = 0.001) in the 100% HED condition (Table 4). An increase in satiety responsiveness score by 1 unit (i.e., child is more satiety responsive) when age was kept constant, decreased HED intake by 15.4 g (*p* = 0.042) on offering the 100% HED condition.

#### 3.5.2. Total Meal Energy Intakes, Demographics and Eating Traits

Positive correlations were found between child age and total meal energy intake (kcal) in both the 100% (*r* = 0.456, *p* = 0.002) and 60% conditions (*r* = 0.363, *p* = 0.017). No correlations were found between total meal energy intakes across both portion conditions and child BMI (100% HED condition *p* = 0.513, 60% condition *p* = 0.532). An inverse correlation was evident between satiety responsiveness and total meal energy intake in the 100% condition (*r* = −0.436, *p* = 0.006) but not the 60% condition (*r* = −0.311, *p* = 0.054) (Figure 5).

Linear regression models were constructed with child age and satiety responsiveness scores as predictors of total meal energy intake (kcal) within each portion size condition (100%, 60%). In the 60% HED portion, child age significantly contributed to the model with age accounting for 17% of the variance in HED intake (*R*^2^ = 0.169, F = 7.55, *p* = 0.009).

Analysis revealed a strong model that both age and satiety responsiveness significantly predicted total meal energy intake in the 100% HED condition (*R*^2^ = 0.36, F = 9.9, *p* < 0.001) with age accounting for 27% of the variance and satiety responsiveness accounting for 9%. An increase in age by 1 year predicted an increased intake from total meal by 76.3 kcal (*p* = 0.004) in the 100% HED condition (Table 5). An increase in satiety responsiveness score by 1 unit (i.e., child is more satiety responsive) when age was kept constant, decreased energy intake by 63.6 kcal (*p* = 0.031) on offering the 100% HED condition.

#### 3.5.3. LED Vegetable Intakes, Demographics and Eating Traits

Controlling for age, an inverse correlation was observed between mean LED intake and child food neophobia score (*r* = −0.637, *p* < 0.001). A child’s LED intake was negatively correlated with child food fussiness (*r* = −0.610, *p* < 0.001) and positively correlated with enjoyment of food (*r* = 0.447, *p* = 0.003), encouraging balance and variety (*r* = 0.490, *p* = 0.001), involvement (*r* = 0.396, *p* = 0.010), teaching nutrition (*r* = 0.337, *p* = 0.031), and pressure (*r* = 0.369, *p* = 0.018). A stepwise linear regression model was explored using the child eating behaviors and parental feeding practices identified above to predict LED intake (Table 6). Analysis revealed that only child food neophobia score significantly contributed to the model (*R*^2^ = 0.45, F = 31.00, *p* < 0.001) with food neophobia accounting for 45% of the variance in LED intake. An increase in child’s food neophobia score by 1 unit (i.e., child is more food neophobic) decreased LED vegetable intake by 4.1 g (*p* < 0.001).

## 4. Discussion

This is the first study to investigate the effect of downsizing a HED main component of a meal paired with LED vegetable variety on pre-school children’s food and energy intake during a lunch meal. The results show a significant effect of downsizing using a unit food on reducing intake of the HED meal item and total meal intake in pre-school children in a nursery setting. The amount of HED sandwich consumed decreased by 21% following a reduction in portion from 100% to 60%, without a compensatory increase in food intake from other meal components. Offering a variety of vegetables as a LED side within the meal increased vegetable intake compared with a single vegetable and moved vegetable intake towards the recommended 40g portion for children in both PS conditions (100%, 60%). Downsizing and variety were effective strategies individually for altering pre-schoolers’ intakes of HED and LED meal items, however, using variety to offset HED downsizing was not supported in the present study.

Providing an age-appropriate portion, compared with the larger portion, to pre-school aged children resulted in significantly lower consumption of the HED component of a meal. These findings complement existing evidence suggesting that the PSE is apparent within children [5,8,52], and also highlight that downsizing a liked HED main-component of a lunchtime meal can be achieved without a compensatory increase in intake from other foods, namely a highly-palatable HED dessert. Our study did not measure dietary intake following the lunch-time meal, however, Rolls et al. [15] found that reducing the portion size and ED of meals and snacks in young women resulted in sustained lower energy intakes over a 2-day period. This evidence along with the lack of compensatory intake observed here suggests the importance of portion control in young children. Age-appropriate portion sizes, particularly those for HED foods, can be learned both by children and their caregivers, and ensure appropriate energy intakes. If retained over time, these learned portion practices may protect against the increasingly obesogenic environment [53], where large portions have become the norm.

Our findings show that the PSE is apparent in children consuming unit-based foods, and supports recent systematic review evidence demonstrating the PSE in both unit and amorphous (foods without a distinct shape or form) foods in children aged 2–12 years [54]. Visual cues, such as the shape of food and how it is presented on a plate, together with social norms, contribute an important role in how much an individual consumes [55,56,57]. Portion size norms may be determined both by the amount served, such as a pre-packaged portion of a snack, but also the number of food items served. Geier et al. [58] similarly concluded that ‘unit bias’, an appropriate number to eat when presented with a food, exists. However, the age at which portion size norms and unit bias develop is currently unknown. In the current study, the number of units across portion size conditions was held constant. It is possible that our pre-schoolers’ may have developed a learned portion norm driven by the number of food units normally served to them. Future research is required to explore and understand how and when a portion norm and unit bias develops. Furthermore, based on the evidence that suggests children prefer more variety in colour and number of items when presented with food on a plate compared to adults [59], future research should consider that the complex role of visual cues, such as unit numbers, shapes, and colours may differ across age groups and individually or combine to impact on acceptance of a food or meal.

Contrary to previous findings [7,8], our data showed no increase in accompanying nutrient-rich LED vegetable intake when reduced quantities of a HED main were consumed during a meal. Offering a variety of vegetables compared with a single vegetable option did increase vegetable intake in the current study, supporting previous work [27], however this variety effect did not differ across portion conditions and thus did not offset the downsizing of the HED food in our sample of pre-school age children. Nevertheless, exposure to a variety of 3 different vegetables concurrently increased vegetable intake by an average of 7.7 g (19% of recommended 40 g portion) compared with the mean single vegetable condition, resulting in intakes close to the recommended portion for children of this age [60]. These encouraging findings support the role of vegetable variety on vegetable intake in pre-school aged children [27,30]. Our findings additionally showed that children consumed similar quantities of vegetables from each of the three types of vegetables when offered the variety condition. Thus, offering a variety of vegetables during a meal not only increases the potential for children to achieve their recommended vegetable portion but additionally provides an opportunity for gaining the benefit of exposure to different flavors, and importantly, different nutrients. Providing children with variety on their plate may encourage acceptance of a meal [59,61] and repeated exposure to less liked foods may increase future acceptance and consumption [62,63,64]. The pairing of different foods on a plate has recently been explored in school-aged children [20] and highlights the role of interactions between flavors and textures during a meal. In our study, the increased variety of food presented on the child’s plate, in the form of a variety of LED vegetables alongside the HED sandwich, would provide sensory variety not only in the form of varying textures and flavors but also visual cues, known to play a role on food intakes and acceptance [65].

Previous studies conducted in children have shown that individual factors can influence the effect of portion size on intake [5,9,14,66]. For example, child’s weight status has been shown to influence the child’s response to larger food portions, with overweight children showing greater increases in intake from large portions compared to their non-overweight counterparts [8,67]. However, no association between HED intake and child’s BMI was found in this study and others [4,5,9]. Our data suggest that a child’s response to an increased portion of familiar and well-liked HED food can be moderated by their individual difference, such as, satiety responsiveness, supporting previous evidence [9,14]. A weak ability to control intake of a HED food when offered a large portion in children with lower satiety responsiveness reiterates the need for parents to implement portion control strategies particularly for HED foods to avoid regular over-consumption and excessive energy intake. Findings from the current study also confirm the role of child neophobia on predicting vegetable intake [44]. The complex roles of heritability [46,68] and family environment [69,70] on eating behaviors together with the present findings, suggest that multiple factors interact to influence a child’s susceptibility to portion size and acceptance of foods offered.

The design, including the natural childcare setting and use of familiar and commonly consumed lunch-time foods [36,38], are strengths of this study. Furthermore, incorporating a popular and typical unit-based HED main food in the meal, and controlling the unit numbers of this food strengthens the evidence of the PSE to include unit as well as amorphous type foods [54], and highlights the potential role of unit bias on pre-school aged children’s eating practices. A limitation of the present study was the lack of a dietary follow-up to monitor intakes beyond a single meal. Future research investigating the effect of food portion downsizing across a longer period is required in young children to investigate sustained effects. The present study was conducted across a period of 21 months and thus traversed the seasons. It is possible that a child’s preference for consuming vegetables varied across the seasons, however to our knowledge there is no literature to support this assumption. The sample size of the present study was similar to previous literature [7,14,35,71], however the final sample included for analysis fell short of the target sample set for this study. Our findings may not be generalizable due to the homogeneous population in terms of socio-economic status and cultural background and cannot be generalized to those eating environments where a family style mealtime is employed where children self-serve their own portions of food. Further research is warranted to investigate the interaction of downsizing and variety in a self-serving setting for young children.

## 5. Conclusions

The findings of this study demonstrate that downsizing the portion of a unit-based HED component of a lunch-time meal can be used as an effective strategy to reduce HED food intake without a compensatory change in intake of other foods, including dessert, in pre-school aged children. An opportunity for promoting children’s vegetable intake to reach recommended portions can be achieved by offering a variety of vegetables at a meal, during an age when food neophobia and eating habits and behaviors are developing [72,73]. However, the results of this study suggest that vegetable variety did not offset the downsizing of the HED component of the meal per se in this age group. Downsizing and variety are simple, effective strategies that can be individually employed by parents and those working in childcare settings to achieve appropriate portion sizes and increase vegetable consumption in children. Providing knowledge on the role of individual characteristics, such as satiety responsiveness and food neophobia, can also assist caregivers in the understanding of how these factors influence child eating.

## Figures and Tables

**Figure 1 nutrients-10-01879-f001:**
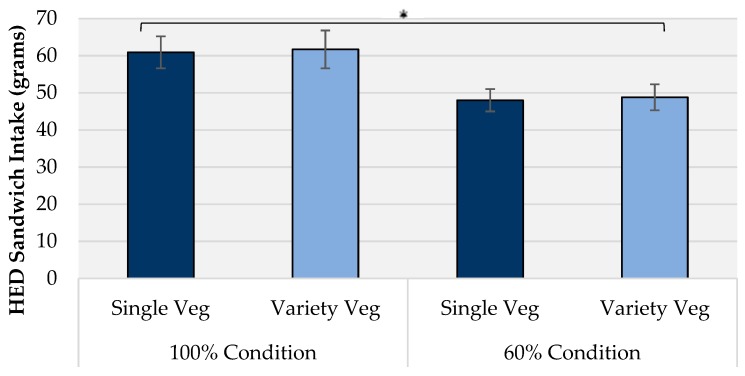
Mean (± SEM) intakes of sandwich at a lunch meal across both HED portion sizes by vegetable condition. * denotes a significant effect of portion size at *p* < 0.05.

**Figure 2 nutrients-10-01879-f002:**
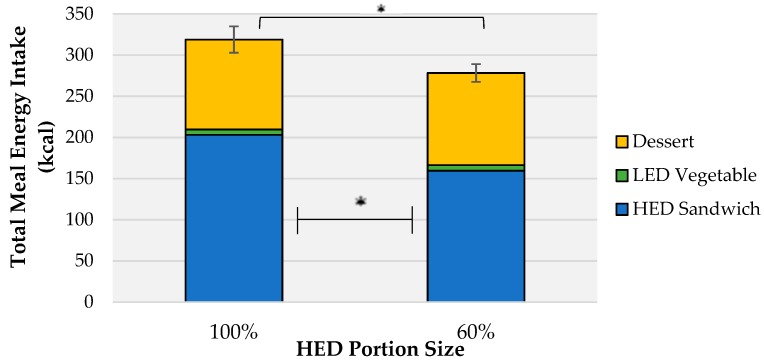
Mean (± SEM) intakes (kcal) of lunch components by HED portion size. Error bars show SEM for total meal intake *denotes a significant effect of portion size condition on at *p* < 0.05.

**Figure 3 nutrients-10-01879-f003:**
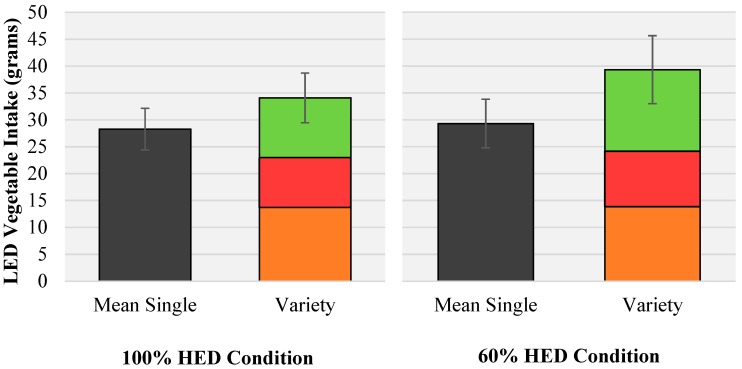
Mean (± SEM) intakes of LED vegetables at a lunch meal across both HED portion sizes by vegetable condition. A significant main effect of vegetable condition was observed at *p* < 0.05. Within each variety condition the mean consumption of each individual vegetable type (carrot (orange), cherry tomato (red) and cucumber (green) has been shown.

**Figure 4 nutrients-10-01879-f004:**
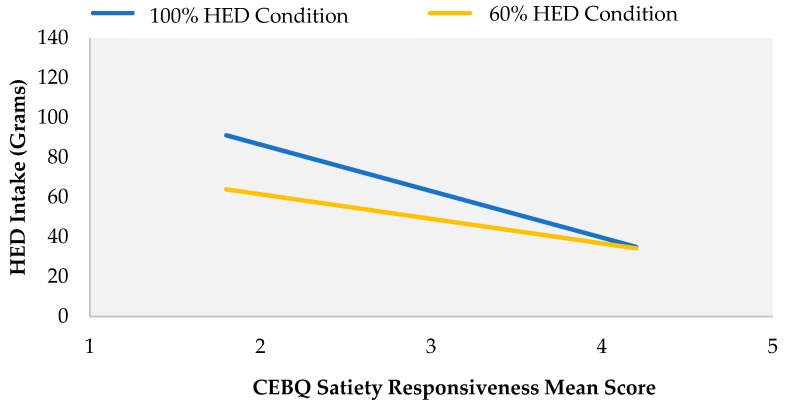
Effect of parental ratings of child satiety responsiveness on HED intake (g) by portion condition.

**Figure 5 nutrients-10-01879-f005:**
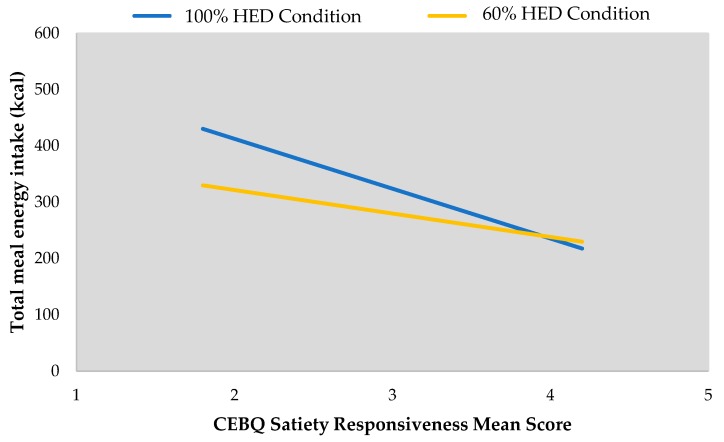
Effect of parental ratings of child satiety responsiveness on total energy intake (kcal) by portion condition.

**Table 1 nutrients-10-01879-t001:** Experimental design.

	Experimental Conditions (weeks)								
		Block 1				Block 2			
Manipulation	1	2	3	4	5	6	7	8	9
Portion Size	Familiarization Session	100%	100%	100%	100%	60%	60%	60%	60%
Vegetable option		SingleA	SingleB	SingleC	VarietyA+B+C	SingleA	SingleB	SingleC	VarietyA+B+C

A = cucumber, B = cherry tomatoes, C = carrot.

**Table 2 nutrients-10-01879-t002:** Characteristics of the test meal provided at lunch.

	100% Portion			60% Portion		
	Weight (g)	Energy (kcal)	Energy Density (kcal/g)	Weight (g)	Energy (kcal)	Energy Density (kcal/g)
Cheese sandwich ^1^	117	368	3.2	70	221	3.2
Vegetables	120	17–26 ^4^	0.1–0.2	120	17–26 ^4^	0.1–0.2
Grapes	40	25	0.6	40	25	0.6
Yogurt ^2^	120	113	0.9	120	113	0.9
**Total Meal** ^3^	397	523–532 ^4^	1.3	350	376–385 ^4^	1.1

^1^ Kingsmill 50/50 © no crust bread, Morrisons brand sunflower spread and medium cheddar cheese; ^2^ Ski^®^ smooth yogurt; ^3^ recommended total energy intake from lunch meal is 371–513kcal; ^4^ dependent on vegetable selection.

**Table 3 nutrients-10-01879-t003:** Characteristics of children participating in study.

	All (*n* = 43)		Girls (*n* = 23)		Boys (*n* = 20)	
	Mean ± SEM	Range	Mean ± SEM	Range	Mean ± SEM	Range
Age (years)	3.9 ± 0.57	3.0–4.9	3.9 ± 0.12	3.0–4.8	4.0 ± 0.13	33.2–4.9
BMI (Kg/m^2^)	16.5 ± 1.33	14.0–19.5	16.9 ± 0.31	14.0–19.5	16.0 ± 0.21	14.5–17.6
% with overweight *	25.6		47.8		0	

*Age and sex specific classification [50,51].

**Table 4 nutrients-10-01879-t004:** 100% HED intake regression model.

	B	SE_B_	ß	*p*
**Step 1**				
Constant	−51.83	25.93		0.053
Age (years)	28.25	6.45	0.59	<0.001
**Step 2**				
Constant	11.70	39.02		0.766
Age (years)	24.33	6.45	0.50	0.001
CEBQ Satiety Responsiveness	−15.42	7.31	−0.28	0.042

Note: *R*^2^ = 0.34 for Step 1; ∆*R*^2^ = 0.07 for Step 2.

**Table 5 nutrients-10-01879-t005:** 100% total meal energy intake regression model.

	B	SE_B_	ß	*p*
**Step 1**				
Constant	−53.20	101.38		0.603
Age (years)	92.44	25.23	0.52	0.003
**Step 2**				
Constant	208.70	151.51		0.177
Age (years)	76.26	25.03	0.43	0.004
CEBQ Satiety Responsiveness	−63.58	28.40	−0.31	0.031

Note: *R*^2^ = 0.27 for Step 1; ∆*R*^2^ = 0.09 for Step 2.

**Table 6 nutrients-10-01879-t006:** Mean LED intake regression model.

	B	SE_B_	ß	*p*
Step 1				
Constant	92.41	11.66		<0.001
Child Food Neophobia Score	−4.08	0.73	−0.67	<0.001

Note: *R*^2^ = 0.45.

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
