# Peer review of "Can Reduced Intake Associated with Downsizing a High Energy Dense Meal Item be Offset by Increased Vegetable Variety in 3–5-year-old Children?"

_nutrients, 2018, doi:10.3390/nu10121879_

Reviewer 1 Report

Comment on Nutrients MS 387807 “Can reduced intake associated with downsizing the portion size of a high energy dense meal item be offset by increased vegetable variety in pre-school children?”

 General comment: The study aimed at testing the effect of reducing the portion size of a high energy dense item on lunch intake and testing whether offering a variety of vegetables increased vegetable intake and offset reduced energy intake associated with the reduced portion size.  Overall, the study appears to be well-designed and well-analyzed.  The findings should be useful to some of Nutrients’ readers.  Below are some comments provided for clarity and completeness.

Specific comment:

Please succinctly characterize the study sample in the title.  “Pre-school children” is too broad.

Line 45 and Results – please clearly identify the findings on offsetting effect in the Results section.  As is, neither 3.3 nor 3.4 seems to directly report the findings.

Line 79 – is this study really a “cross-over” design? Where did the “cross” take place?  The description seems to indicate two possible blocks of conditions and subjects were randomly assigned either a 100%-60% sequence or a 60%-100% sequence.  Is this correct?  If yes, this is not a cross-over design.  If this is not correct, please add clarification about the exact design.

Line 102 – were the single vegetables also raw? Please clarify.

Line 128 – please indicate the number of nurseries included.  Since there were nine of them, could there have been any cluster effects due to differences between nurseries?

Line 171-172 – please explain why the Bonferroni adjustment was only applied to “outcomes with a significant effect”?  Whether outcomes in multiple comparisons are significant should be based on adjusted test statistics to begin with.

Line 185 – were all children tested at the same time in a year?  If not, could there have been any seasonable effects on the outcomes, e.g., children may prefer vegetables less in the colder months than in the warmer months?

Figure 3 – please check to make sure Figure 3 is correctly positioned.  The current figure does not appear to match the text.      

Author Response

1.      Please succinctly characterize the study sample in the title.  “Pre-school children” is too broad.

Response: Thank you for your comment.  We have revised the title to make clear the age of the children and have also shortened  this a little: ‘Can reduced intake associated with downsizing a high energy dense meal item be offset by increased vegetable variety in 3-5-year old children?’. We have additionally stated 3-5-year-old pre-schoolers in the abstract to define the group.

2.      Line 45 and Results – please clearly identify the findings on offsetting effect in the Results section.  As is, neither 3.3 nor 3.4 seems to directly report the findings.

Response: Thank you for this comment.  We use the term offset to interpret the data in the discussion, but to make this clear in the results we have added the following to the results descriptions (section 3.2) “There was no  significant interaction effect of portion size and vegetable condition (p=0.995) indicating vegetable variety did not offset portion downsizing.

3.      Line 79 – is this study really a “cross-over” design? Where did the “cross” take place?  The description seems to indicate two possible blocks of conditions and subjects were randomly assigned either a 100%-60% sequence or a 60%-100% sequence.  Is this correct?  If yes, this is not a cross-over design.  If this is not correct, please add clarification about the exact design.

Response: Thank you for your comment. We have corrected this and removed the “cross-over” terminology as the participants were randomly assigned one of two sequences.

4.      Line 102 – were the single vegetables also raw? Please clarify.

Response: Thank you for this comment. We have described in lines 104 that the vegetables both single and variety were raw in their presentation.

5.      Line 128 – please indicate the number of nurseries included.  Since there were nine of them, could there have been any cluster effects due to differences between nurseries?

Response: Thank you for this comment. We have reworded this sentence to “The number of participating children from each of the 9 nurseries ranged from 2 to 9 children.” We have sought advice from our statistician and have conducted a cluster analysis on the data using the STATA procedure (xtmixed) using their maximum likelihood estimator and  tried to fit the cluster control solution.  The model failed to converge, we suspect that was due to too many groups and too few participants (sample size restrictions). The ICC for nursery (separate from variation within the level of child) is very small (0.0001) which is reflective of a low value for small observational studies such as ours and suggests there is little variation across the nurseries.  This is also consistent with how the nurseries hosted, but did not conduct, the testing – the testing was conducted by the same researchers across nurseries.

6.      Line 171-172 – please explain why the Bonferroni adjustment was only applied to “outcomes with a significant effect”?  Whether outcomes in multiple comparisons are significant should be based on adjusted test statistics to begin with.

Response: Thank you for this comment, we have clarified this statement as follows “The Bonferroni method was used to adjust significance levels for multiple pairwise comparisons between means”.

7.      Line 185 – were all children tested at the same time in a year?  If not, could there have been any seasonable effects on the outcomes, e.g., children may prefer vegetables less in the colder months than in the warmer months?

Response: The children were tested throughout the year. Thank you for highlighting this point. We are not aware of any literature regarding vegetable preferences changing across the seasons. We had added this point within the discussion lines 397-400 – The present study was conducted across a period of 21 months and thus traversed the seasons. It is possible that a child’s preference for consuming vegetables varied across the seasons, however to our knowledge there is no literature to support this assumption.”

8.      Figure 3 – please check to make sure Figure 3 is correctly positioned.  The current figure does not appear to match the text.  

Response: Checked and corrected thank you

Reviewer 2 Report

This is a well written manuscript, with a sound design. Strengths include the repeated measures design to assess two main factors. The research question is important to reiterate to care-providers to avoid excess serving sizes and increase vegetable variety for young children.

I have a few comments.

Methods

Pearson's correlations are not the best choice for age or BMI with intake, as age (range 3-5) and BMI were likely not normal. Can the authors at least confirm that a linear bivariate relationship existed for variables where Pearson's correlations were used?

Results

Child BMI should be age- and sex-adjusted using accepted criteria (e.g., WHO) and reported as z scores or percentiles.

Why are not all demographics reported in Table 1? Why just in the text?

Figure 2: The between group difference was about 42 kCals over the entire meal. How meaningful is that difference?

The yogurt serving appears to have been 120 g. Was that one, single-serving container of yogurt? How many children ate the entire container? Perhaps all children ate all of the yogurt, thus there was no room to overcompensate for the lower meal serving size.

Figure 3: The bars presented are means +/- SEM. The bars overlap, suggesting that the means are not different at the 68% confidence level (ie, mean +/- 1*SEM is the 68% confidence interval). The figure notes there is a difference at the P<0.05 level. Perhaps the error bars on the figure are incorrect?

Why are the analyses in lines 234-239 included? How do these analyses inform the primary research question? I think they are somewhat distracting.

Section 3.5.1 and Figure 4: I am not sure why the analyses related to satiety responsiveness and HED intake are included. They don't add novel information, and they are distracting from the primary research aims. Also, the fact that food intake increases with age appears obvious and makes the reader question why analyses weren't adjusted for the more important predictor of estimated expected energy intake. (To note, the primary analyses are fine looking at total intake because they were repeated measures.) 

Also, why were these analyses (3.5.1) conducted just on HED sandwich intake and not the entire meal intake? Wouldn't it be more useful to understand if those with low satiety responsiveness compensate less over the entire meal?

Conversely, the analyses assessing predictors of variety of vegetable intake, 3.5.2, are more relevant and suggest that many children will benefit from increased vegetable variety--if food neophobia is not a significant factor for children.

I appreciate the limitation that these findings may not be relevant for those serving "family-style" in child care settings.       

Author Response

Pearson's correlations are not the best choice for age or BMI with intake, as age (range 3-5) and BMI were likely not normal.  Can the authors at least confirm that a linear bivariate relationship existed for variables where Pearson's correlations were used?

Response: Thank you we have reworded this to clarify  - Pearson’s correlation was used for linear bivariate relationships to explore associations between mean intakes, child age and BMI, eating traits and parental feeding practices.”

Child BMI should be age- and sex-adjusted using accepted criteria (e.g., WHO)  and reported as z scores or percentiles.

Response: Thank you for this comment. We have intentionally used the International Obesity Task Force (IOTF) categorisation which adjusts for age and gender of the child.  These are internationally acceptable cut-off points for child obesity and overweight which allow trends in childhood obesity worldwide to be compared and which was our rationale for use.  We would please like to retain our use of these cut-offs and we have included an additional reference to the footnote at Table 3, Cole & Lobstein (2012), which provides the most up to date validation of these.

Why are not all demographics reported in Table 1? Why just in the text?

Response: Thank you for this comment. We have added extra columns into Table 3 to show the data for the overall sample.

Figure 2: The between group difference was about 42 kCals over the entire meal. How meaningful is that difference?

Response: Thank you for this comment we have added the following to line 217-218 where Figure 2 data is described - This difference equates to 9-12% of the total energy intake recommended for a child of this age at a lunchtime meal.”

The yogurt serving appears to have been 120 g. Was that one, single-serving container of yogurt? How many children ate the entire container? Perhaps all children ate all of the yogurt, thus there was no room to overcompensate for the lower meal serving size.

Response: The yogurt was 120g served as one container. This was not manipulated in the study design and was included alongside the grapes to ensure we met the recommended meal provision for children in a nursery setting. The number of children who completely finish the yogurt was low (approx. 3-4 in each condition) and mean yogurt intake ranged from 92g to 102g across the 8 conditions.

Figure 3: The bars presented are means +/- SEM. The bars overlap, suggesting that  the means are not different at the 68% confidence level (ie, mean +/-  1*SEM is the 68% confidence interval). The figure notes there is a  difference at the P<0.05 level. Perhaps the error bars on the figure  are incorrect?

Response: Thank you for this comment.  We have double checked, and the data presented in figure 3 are correct however, we have removed the * significance in both PS conditions on this figure and changed wording in the figure description to clarify that there is a significant main effect of vegetable condition on vegetable intake and additionally added text to the results to clarify - it now reads Offering a variety of vegetables at the lunch meal resulted in a higher vegetable intake (37.2 ± 5.2g) compared with a mean single vegetable option (29.5 ± 4.1g) without an interaction effect with PS condition (F(1,41) = 1.58, p=0.216).”

Why are the analyses in lines 234-239 included? How do these  analyses inform the primary research question? I think they are somewhat distracting.

Response: Thank you. In our original manuscript, lines 234-239 are data describing correlations between eating traits and child age or BMI (relationships with subject characteristics) which we believe are of interest within this research field and would wish for these to be retained.

Section 3.5.1 and Figure 4: I am not sure why the analyses related to satiety responsiveness and HED intake are included. They don't add novel information, and they are distracting from the primary research aims. Also, the fact that food intake increases with age appears obvious and makes the reader question why analyses weren't adjusted for the more important predictor of estimated expected energy intake. (To note, the primary analyses are fine looking at total intake because they were repeated measures.)  Also, why were these analyses (3.5.1) conducted just on HED sandwich intake and not the entire meal intake? Wouldn't it be more useful to understand if those with low satiety responsiveness compensate less over the entire meal?

Response: Thank you, we agree with the suggestion that the analysis on total meal energy intake would be useful to understand and have included this additional analysis alongside the analysis on HED intake on pages 9 and 10. We feel the analysis relating to satiety responsiveness in this age group is a novel finding and maintain that this is an important finding for this manuscript; it supports the contention proposed by Kral and Hetherington (2015) that portion size effects are influenced by individual differences.